# Are responders to patient health surveys representative of those invited to participate? An analysis of the Patient-Reported Outcome Measures Pilot from the Australian Orthopaedic Association National Joint Replacement Registry

Ian A. Harris[1,2]*, Kara Cashman[1], Michelle Lorimer[1], Yi Peng[1], Ilana Ackerman[3], Emma Heath[4], Stephen E. Graves[1]

1 Australian Orthopaedic Association National Joint Replacement Registry (AOANJRR), Clinical and Health Sciences, University of South Australia, Adelaide, South Australia, Australia, 2 Ingham Institute for Applied Medical Research, South Western Sydney Clinical School, UNSW Sydney, Liverpool Hospital, Liverpool, New South Wales, Australia, 3 School of Public Health and Epidemiology, Monash University, Melbourne, Victoria, Australia, 4 South Australian Health and Medical Research Institute (SAHMRI), Adelaide, South Australia, Australia

* ianharris@unsw.edu.au

## Abstract

### Background

Patient-reported outcome measures (PROMs) are commonly used to evaluate surgical outcome in patients undergoing joint replacement surgery, however routine collection from the target population is often incomplete. Representative samples are required to allow inference from the sample to the population. Although higher capture rates are desired, the extent to which this improves the representativeness of the sample is not known. We aimed to measure the representativeness of data collected using an electronic PROMs capture system with or without telephone call follow up, and any differences in PROMS reporting between electronic and telephone call follow up.

### Methods

Data from a pilot PROMs program within a large national joint replacement registry were examined. Telephone call follow up was used for people that failed to respond electronically. Data were collected pre-operatively and at 6 months post-operatively. Responding groups (either electronic only or electronic plus telephone call follow up) were compared to non-responders based on patient characteristics (joint replaced, bilaterality, age, sex, American Society of Anesthesiologist (ASA) score and Body Mass Index (BMI)) using chi squared test or ANOVA, and PROMs for the two responder groups were compared using generalised linear models adjusted for age and sex. The analysis was restricted to those undergoing primary elective hip, knee or shoulder replacement for osteoarthritis.

**Data Availability Statement:** Data cannot be shared publicly because of legislative restrictions on the use of registry data from the Australian Orthopaedic Association National Joint Replacement Registry (AOANJRR). Data are available from the AOANJRR for those who meet the criteria for access. External access to and use of de-identified AOANJRR data is permitted but must be in accordance with AOANJRR policies (Ref No POL.S3.3, S3.4, S3.5) available on the registry website: https://aoanjrr.sahmri.com/policies. Requests for data use can be made by contacting the AOANJRR Manager: Cindy Turner Manager AOANJRR Telephone: +618 8128 4284 Email: cturner@aoanjrr.org.au.

**Funding:** The authors received no specific funding for this work.

**Competing interests:** The authors have declared that no competing interests exist.

## Results

Pre-operatively, 73.2% of patients responded electronically and telephone follow-up of non-responders increased this to 91.4%. Pre-operatively, patients responding electronically, compared to all others, were on average younger, more likely to be female, and healthier (lower ASA score). Similar differences were found when telephone follow up was included in the responding group. There were little (if any) differences in the post-operative comparisons, where electronic responders were on average one year younger and were more likely to have a lower ASA score compared to those not responding electronically, but there was no significant difference in sex or BMI. PROMs were similar between those reporting electronically and those reporting by telephone.

## Conclusion

Patients undergoing total joint replacement who provide direct electronic PROMs data are younger, healthier and more likely to be female than non-responders, but these differences are small, particularly for post-operative data collection. The addition of telephone call follow up to electronic contact does not provide a more representative sample. Electronic-only follow up of patients undergoing joint replacement provides a satisfactory representation of the population invited to participate.

## Introduction

Patient reported outcome measures (PROMs) of health status are commonly recorded pre- and post-operatively in people undergoing joint replacement surgery as a measure of surgical thresholds and treatment effects for these common and resource intensive procedures. However, unlike registries that commonly have near-complete coverage of all procedures, PROMs collection is rarely complete, being limited by resources and patient responsiveness. Registry-based PROMs collection in joint replacement surgery has coverage rates rarely higher than 80%, and often less than 50% [1,2]. A 60% threshold has been suggested for completeness in PROMs collection [3], but with any threshold, it is important to know the representativeness of the sample so that conclusions based on the sample can be applied to the population. We consider it more important to understand the representativeness of a sample than the size of the sample or completeness. For example, data from a 50% sample may be considered meaningful if the differences between the sample and the population are understood, whereas an 80% sample may provide misleading information if the sample is unrepresentative.

The Australian Orthopaedic Association National Joint Replacement Registry (AOANJRR) piloted a PROMs program, targeting all patients undergoing elective hip, knee or shoulder replacement from participating institutions. Due to the inefficiencies associated with using paper forms (either directly or by mail) within a national registry, the AOANJRR PROMs program uses direct electronic data capture. Data capture by telephone call using an interviewer who directly entered data electronically was also used for the pilot stage of the program, but only for patients who did not respond to direct electronic data entry following text message and email prompts for completion.

This study aims to answer the following questions: 1) are patients for whom PROMs data were captured directly (electronically) different to those who did not provide the data electronically (i.e. all others); 2) does adding telephone call follow up to those responding electronically

improve representativeness; and 3) are patient-reported outcomes different between those who provide direct electronic data entry and those who provide information via telephone call after not responding electronically?

## Methods

Between 30 July 2018 and 28 January 2020, the AOANJRR conducted the first stage of a PROMs pilot study, collecting PROMs data from patients pre-operatively and at six months post-surgery from 43 institutions across Australia, including metropolitan and regional, and private and public hospitals from all states and one territory. The analysis includes procedures registered (pre-op analysis) or performed (post-op analysis) between 30 July 2018 and 29 May 2019, to allow 8 months for follow up. The study was nested within the AOANJRR, a national registry that validates more than 97.8% of all joint replacement procedures for all hospitals (approximately 320) performing joint replacement surgery in Australia [4].

The following Australian ethics committees approved the pilot program from which these data were drawn: University of South Australia HREC (200890), Sydney Local Health District Ethics Review Committee (RPAH Zone, HREC/18/RPAH/90), Calvary Health Care Adelaide HREC (18-CHREC-F004), Mater Misericordiae Ltd HREC (HREC/18/MHS/45), St Vincent's Health and Aged Care HREC (HREC 18/14), University of Tasmania HREC (H0017292), Calvary Health Care Tasmania HREC (010418), St John of God HREC (1408), Calvary Health Care (ACT)(25–2018). Consent was not obtained for the analyses used in this report as data were analyzed anonymously. Researchers accessed data on 18 May 2020, after the data were anonymized.

Data collection involved initial (pre-operative) patient data capture at hospital pre-admission clinics or private surgeon clinics using electronic-only methods which had the capacity to be conducted on multiple devices including smart phone, tablet or computer. Patients unable to complete data entry at initial contact were registered in the system and contacted electronically via email or text message two and five days after initial registration to allow direct data entry at their convenience. Post-operative collection involved direct electronic contact via email or text message links directing the patient to the online survey, which were sent 166 and 180 days after the patients' procedure date. Non-responders or those without any electronic contacts including those with only home telephone numbers (both pre-operative and post-operative) were contacted by telephone and asked to complete the survey by telephone.

The analysis was restricted to primary elective procedures undertaken for osteoarthritis. The denominator used for the main analyses (the reference population) was all patients who were registered for the PROMs pilot program that were matched to routine AOANJRR data, because many patients who were registered may not have proceeded to surgery within the study period. Using known procedures and registered patients tests the 'within-system' representativeness by restricting the analysis to patients who were given the opportunity to respond. The comparison of telephone to electronic follow up was further restricted to those who had an email or mobile telephone number, again to compare the responsiveness in similar patient populations.

Information on the population included data routinely collected by the AOANJRR; age, gender, Body Mass Index (BMI), American Society of Anesthesiologists (ASA) physical status classification [5], unilateral/bilateral and approach (for hip replacement). Other demographic information (e.g., education and ethnicity) was not available for analysis. PROMs data include the Oxford Hip Score [6] (OHS), Oxford Knee Score [7] (OKS), EQ-5D-5L [8] Utility Index (using Australian preference weights) and Visual Analogue Scale (VAS), low back pain, affected joint pain, expected (post-operative) pain and function, the Hip injury and

Osteoarthritis Outcome Score, 12 item [9] (HOOS-12) and the Knee injury and Osteoarthritis Outcome Score, 12 item [10] (KOOS-12), the latter two scores providing a summary score and domain scores for Pain, Function and Quality of Life.

Categorical data (proportions) were compared using chi squared tests. Continuous demographic data were compared using analysis of variance. Differences in pre-and post-operative PROMs responses between groups were compared using general linear models adjusting for age and sex. The critical value chosen to reject the alternative hypothesis was 0.05.

## Results

Over the study period, 6,224 patients at 43 participating institutions having electronic contact information and primary THA, TKA or TSA for osteoarthritis were registered into the PROMs pilot study.

Of the 6,224 patients registered into the PROMs system, 417 patients had their pre-operative PROMs data collected directly by separate hospital systems, external to the electronic data capture system, and were excluded from pre-op analyses. A further 48 patients were excluded from pre-op analyses due to pre-op follow up phone calls ceasing. For post-op analyses, of the 6,224 patients registered, 24 died and 55 opted out prior to completing their post-op PROMs and were excluded from post-op analyses. A further 573 patients had their procedure after 29 May 2019 and were unable to be followed up by telephone due to post-op follow up phone calls ceasing and are excluded from post-op analyses.

For pre-op analyses, a total of 5,759 patients with a mobile phone and/or email address listed were matched to 6,095 primary hip, knee or shoulder replacement procedures performed for osteoarthritis over the same period and are included in the pre-op analyses. For post-op analyses, a total of 5,572 patients were matched to 5,892 primary hip, knee or shoulder replacement procedures performed for osteoarthritis and are included in the post-op analyses. For patients with multiple procedures, only the first procedure was included in the demographic analyses.

### 1. Electronic responders versus electronic non-responders

Of the 5,759 registered patients, 4,213 (73.2%) completed pre-operative PROMs data electronically, the remainder ("electronic non-responders") were either followed up by telephone or not followed.

A comparison of those responding to pre-operative electronic data collection to electronic non-responders is provided in Table 1. There was no difference in the type of joint replacement between the groups, however, patients having bilateral procedures were more likely to respond. On average, responders were one year younger and more likely to be female. The average difference in BMI between responders and non-responders (0.3 kg/m$^2$) was small.

The post-operative comparisons are provided in Table 1 and show that responders were, on average, one year younger and more likely to be healthy (lower ASA class) but there was no significant difference for BMI, sex or joint type.

### 2. All responders (electronic plus telephone) versus non-responders

The addition of telephone call follow up for patients not responding (or not able to respond) to electronic data entry increased pre-operative data completeness from 73.2% to 91.4%. The differences between all responders (electronic and telephone) and non-responders are provided in Table 2. There were no significant between group differences in the pre-operative or post-operative characteristics of all responders compared to non-responders.

**Table 1. Demographic and clinical comparison of patients providing electronic data entry ("responders") to those not providing data electronically ("electronic non-responders"\*).**

| Patient Characteristic | | Pre-Operative | | | | Post-Operative | | | |
|---|---|---|---|---|---|---|---|---|---|
| | | Total | Electronic Responders | Electronic Non-Responders* | P Value | Total | Electronic Responders | Electronic Non-Responders* | P Value |
| Age | n | 5759 | 4213 | 1546 | | 5572 | 2734 | 2838 | |
| | mean (SD) | 66.65 (9.38) | 66.33 (9.23) | 67.52 (9.71) | <0.0001 | 66.63 (9.43) | 66.20 (8.84) | 67.04 (9.95) | 0.0009 |
| BMI | n | 5692 | 4170 | 1522 | | 5508 | 2712 | 2796 | |
| | mean (SD) | 31.59 (6.56) | 31.67 (6.62) | 31.37 (6.36) | 0.12 | 31.55 (6.57) | 31.39 (6.60) | 31.70 (6.54) | 0.08 |
| Gender | Female | 3129 (54%) | 2329 (55%) | 800 (52%) | 0.02 | 3031 (54%) | 1483 (54%) | 1548 (55%) | 0.82 |
| | Male | 2630 (46%) | 1884 (45%) | 746 (48%) | | 2541 (46%) | 1251 (46%) | 1290 (45%) | |
| ASA | 1 | 324 (5.6%) | 240 (5.7%) | 84 (5.4%) | 0.15 | 321 (5.8%) | 182 (6.7%) | 139 (4.9%) | <0.0001 |
| | 2 | 3137 (55%) | 2321 (55%) | 816 (53%) | | 3068 (55%) | 1564 (57%) | 1504 (53%) | |
| | 3 | 2235 (39%) | 1611 (38%) | 624 (40%) | | 2125 (38%) | 965 (35%) | 1160 (41%) | |
| | 4 | 48 (0.8%) | 30 (0.7%) | 18 (1.2%) | | 44 (0.8%) | 18 (0.7%) | 26 (0.9%) | |
| Joint | Hip | 2114 (37%) | 1535 (36%) | 579 (37%) | 0.13 | 2084 (37%) | 1058 (39%) | 1026 (36%) | 0.14 |
| | Knee | 3508 (61%) | 2587 (61%) | 921 (60%) | | 3361 (60%) | 1614 (59%) | 1747 (62%) | |
| | Shoulder | 137 (2.4%) | 91 (2.2%) | 46 (3.0%) | | 127 (2.3%) | 62 (2.3%) | 65 (2.3%) | |
| Unilateral/ Bilateral | Unilateral | 5472 (95%) | 3980 (94%) | 1492 (97%) | 0.002 | 5292 (95%) | 2583 (94%) | 2709 (95%) | 0.10 |
| | Bilateral | 287 (5.0%) | 233 (5.5%) | 54 (3.5%) | | 280 (5.0%) | 151 (5.5%) | 129 (4.5%) | |

\* including those who responded via telephone and non-responders.

Characteristics of electronic-only responders, all responders (electronic and telephone) and all patients are provided in Table 3 to quantify the difference in representativeness when telephone responders are added to electronic-only responders (compared to the total group). Statistical tests are not provided as nearly all differences were statistically significant due to the large sample size. There is no more than 1% or one unit (for age and BMI) difference between electronic-only and all responders (electronic plus telephone), except for post-operative ASA class where the addition of telephone follow up increased the response of ASA class 3 patients.

## 3. Patient reported outcomes in those responding electronically versus by telephone

Pre-operative and post-operative patient-reported outcomes for total hip arthroplasty (THA) patients responding electronically versus telephone follow up are provided in Table 4A and 4B. No clinically meaningful or statistically significant differences were identified for the pre-operative PROMs, and only a small difference in post-operative EQ-5D VAS scores was evident.

Comparisons between electronic and telephone responders for total knee arthroplasty (TKA) patients pre- and post-operatively are provided in Table 5A and 5B.

**Table 2. Comparison of patients responding electronically and by telephone compared to non-responders.**

| Patient Characteristic | | Pre-Operative | | | | Post-Operative | | | |
|---|---|---|---|---|---|---|---|---|---|
| | | Total | Responders | Non-Responder | P Value | Total | Responder | Non-Responder | P Value |
| Age | n | 5759 | 5263 | 496 | | 5572 | 4357 | 1215 | |
| | mean (SD) | 66.65 (9.38) | 66.62 (9.28) | 66.95 (10.30) | 0.45 | 66.63 (9.43) | 66.73 (9.21) | 66.29 (10.18) | 0.15 |
| BMI | n | 5692 | 5205 | 487 | | 5508 | 4308 | 1200 | |
| | mean (SD) | 31.59 (6.56) | 31.60 (6.60) | 31.47 (6.12) | 0.68 | 31.55 (6.57) | 31.59 (6.56) | 31.42 (6.63) | 0.43 |
| Gender | Female | 3129 (54%) | 2880 (55%) | 249 (50%) | 0.05 | 3031 (54%) | 2369 (54%) | 662 (54%) | 0.94 |
| | Male | 2630 (46%) | 2383 (45%) | 247 (50%) | | 2541 (46%) | 1988 (46%) | 553 (46%) | |
| ASA | 1 | 324 (5.6%) | 298 (5.7%) | 26 (5.3%) | 0.95 | 321 (5.8%) | 247 (5.7%) | 74 (6.1%) | 0.92 |
| | 2 | 3137 (55%) | 2871 (55%) | 266 (54%) | | 3068 (55%) | 2407 (55%) | 661 (55%) | |
| | 3 | 2235 (39%) | 2038 (39%) | 197 (40%) | | 2125 (38%) | 1659 (38%) | 466 (39%) | |
| | 4 | 48 (0.8%) | 44 (0.8%) | 4 (0.8%) | | 44 (0.8%) | 35 (0.8%) | 9 (0.7%) | |
| Joint | Hip | 2114 (37%) | 1929 (37%) | 185 (37%) | 0.57 | 2084 (37%) | 1636 (38%) | 448 (37%) | 0.74 |
| | Knee | 3508 (61%) | 3212 (61%) | 296 (60%) | | 3361 (60%) | 2619 (60%) | 742 (61%) | |
| | Shoulder | 137 (2.4%) | 122 (2.3%) | 15 (3.0%) | | 127 (2.3%) | 102 (2.3%) | 25 (2.1%) | |
| Unilateral/Bilateral | Unilateral | 5472 (95%) | 4997 (95%) | 475 (96%) | 0.42 | 5292 (95%) | 4147 (95%) | 1145 (94%) | 0.18 |
| | Bilateral | 287 (5.0%) | 266 (5.1%) | 21 (4.2%) | | 280 (5.0%) | 210 (4.8%) | 70 (5.8%) | |

Results for total shoulder arthroplasty (TSA) patients are not shown. For this group, there was one significant difference, whereby patients responding by telephone reported less post-operative joint pain than electronic responders (1.0 versus 3.1, mean difference 2.0, 95%CI 0.9–3.1).

## Discussion

Our findings show that patients undergoing elective joint replacement surgery who are included in a PROMs program using direct, electronic data entry are, on average, younger, more likely to be female and have a lower ASA score than those that do not take part. However, the differences are small, particularly in the post-operative comparisons where there was a 1-year difference in age and a 1% difference in the distribution of sex. When telephone call follow up was added to the responder group, there were very little or no differences in the pre- and post-operative comparison to non-responders. The small differences seen may not be clinically important and statistical significance, where present, likely reflects the large sample size and corresponding statistical power. The representativeness of the samples did not change by more than 1% or one unit (year of age or unit of BMI) when telephone follow up was added to electronic follow up, except that telephone follow up detected more patients in a higher ASA class.

A previous study in elective surgery patients in England (including hip and knee arthroplasty) showed that responding patients were more likely to be female and older (which concurs with our findings) [11]. A recent study of patients included in a hip arthroscopy registry also reported that responding patients were more likely to be female and older, but they included a broader age spectrum (more younger people) [12]. Similar to our study, the differences between responders and non-responders in both these studies were small. A study of shoulder arthroplasty patients, however, showed that female patients were less likely to respond [13].

Comparing PROMs between electronic and telephone responders showed no significant difference for most outcomes. Pain outcomes, however, were often different between these groups, with patients responding electronically reporting higher expected pain than those

**Table 3. Characteristics of electronic-only responders, electronic plus telephone responders and all patients.**

| Patient Characteristic | | Pre-operative | | | Post-operative | | |
|---|---|---|---|---|---|---|---|
| | | Total | Electronic responders | Electronic + telephone responder | Total* | Electronic responders | Electronic + telephone responder |
| Age | n | 5759 | 4213 | 5263 | 5572 | 2734 | 4357 |
| | mean (SD) | 66.65 (9.38) | 66.33 (9.23) | 66.62 (9.28) | 66.63 (9.43) | 66.20 (8.84) | 66.73 (9.21) |
| BMI | n | 5692 | 4170 | 5205 | 5508 | 2712 | 4308 |
| | mean (SD) | 31.59 (6.56) | 31.67 (6.62) | 31.60 (6.60) | 31.55 (6.57) | 31.39 (6.60) | 31.59 (6.56) |
| Gender | Female | 3129 (54%) | 2329 (55%) | 2880 (55%) | 3031 (54%) | 1483 (54%) | 2369 (54%) |
| | Male | 2630 (46%) | 1884 (45%) | 2383 (45%) | 2541 (46%) | 1251 (46%) | 1988 (46%) |
| ASA | 1 | 324 (5.6%) | 240 (5.7%) | 298 (5.7%) | 321 (5.8%) | 182 (6.7%) | 247 (5.7%) |
| | 2 | 3137 (55%) | 2321 (55%) | 2871 (55%) | 3068 (55%) | 1564 (57%) | 2407 (55%) |
| | 3 | 2235 (39%) | 1611 (38%) | 2038 (39%) | 2125 (38%) | 965 (35%) | 1659 (38%) |
| | 4 | 48 (0.8%) | 30 (0.7%) | 44 (0.8%) | 44 (0.8%) | 18 (0.7%) | 35 (0.8%) |
| Joint | Hip | 2114 (37%) | 1535 (36%) | 1929 (37%) | 2084 (37%) | 1058 (39%) | 1636 (38%) |
| | Knee | 3508 (61%) | 2587 (61%) | 3212 (61%) | 3361 (60%) | 1614 (59%) | 2619 (60%) |
| | Shoulder | 137 (2.4%) | 91 (2.2%) | 122 (2.3%) | 127 (2.3%) | 62 (2.3%) | 102 (2.3%) |
| Unilateral/Bilateral | Unilateral | 5472 (95%) | 3980 (94%) | 4997 (95%) | 5292 (95%) | 2583 (94%) | 4147 (95%) |
| | Bilateral | 287 (5.0%) | 233 (5.5%) | 266 (5.1%) | 280 (5.0%) | 151 (5.5%) | 210 (4.8%) |

* Total post-operative number is restricted to those reaching 8 months post-operative.

responding by telephone. Post-operatively, however, while TKR patients responding electronically reported higher pain levels than those responding by telephone, there was no significant difference for THR patients. A similar study recently reported that patients undergoing hip surgery who responded electronically reported significantly but marginally more pain than in non-electronic responders, but no difference in knee or shoulder patients [14]. It should be noted that the differences in PROMs between electronic and telephone responders were small (all were less than clinically important thresholds) and any statistical significance likely reflects the large sample size that permitted the detection of small differences.

The differences in age and ASA score are likely due to lack of resources and individual capacity to respond electronically in older and less healthy patients. Interestingly, when younger orthopaedic patients have been included, they have been reported as having lower electronic response rates [14].

Given that the PROMs tools used in the current study were designed for direct patient entry, it is likely that any discrepancy between direct patient entry and telephone follow up is due to bias in the group responding by telephone, possibly due to features of the interviewer-patient interaction. Previous research has shown no difference between scores reported by telephone and those recorded by direct patient entry (paper based) for the EQ-5D survey or the Oxford hip or knee scores on patients undergoing joint replacement or other orthopaedic

**Table 4. a: Pre-Operative Patient-reported outcomes comparing patients responding directly ("electronic") and those responding by telephone ("telephone") for patients undergoing THA. b: Post-operative patient-reported outcomes comparing patients responding directly ("electronic") and those responding by telephone ("telephone") for patients undergoing THA.**

| PROM | | Electronic | Telephone | Adjusted Difference (95% CI) | P Value |
|---|---|---|---|---|---|
| EQ-5D-5L Utility Index | N | 1555 | 399 | | |
| EQ-5D-5L Utility Index | Mean (SE) | 0.38 (0.01) | 0.37 (0.02) | 0.01 (-0.02, 0.05) | 0.44 |
| EQ-5D VAS | N | 1540 | 399 | | |
| EQ-5D VAS | Mean (SE) | 67.71 (0.51) | 66.89 (0.92) | 0.82 (-1.23, 2.88) | 0.43 |
| Lower Back Pain | N | 1537 | 397 | | |
| Lower Back Pain | Mean (SE) | 4.10 (0.08) | 4.22 (0.16) | -0.12 (-0.46, 0.22) | 0.49 |
| Affected Joint Pain | N | 1513 | 396 | | |
| Affected Joint Pain | Mean (SE) | 6.96 (0.05) | 6.79 (0.11) | 0.17 (-0.06, 0.41) | 0.147 |
| Oxford Hip Score | N | 1518 | 396 | | |
| Oxford Hip Score | Mean (SE) | 20.72 (0.22) | 21.26 (0.46) | -0.54 (-1.54, 0.46) | 0.291 |
| HOOS-12 Pain | N | 1020 | 316 | | |
| HOOS-12 Pain | Mean (SE) | 38.54 (0.55) | 38.86 (1.07) | -0.32 (-2.68, 2.04) | 0.790 |
| HOOS-12 Function | N | 1013 | 314 | | |
| HOOS-12 Function | Mean (SE) | 46.07 (0.62) | 44.07 (1.15) | 2.00 (-0.57, 4.57) | 0.128 |
| HOOS-12 Quality of Life | N | 1007 | 313 | | |
| HOOS-12 Quality of Life | Mean (SE) | 31.48 (0.60) | 29.37 (1.14) | 2.11 (-0.41, 4.63) | 0.101 |
| HOOS-12 Summary | N | 1007 | 313 | | |
| HOOS-12 Summary | Mean (SE) | 38.62 (0.54) | 37.37 (1.02) | 1.25 (-1.01, 3.51) | 0.278 |
| Expected Joint Pain | N | 1508 | 395 | | |
| Expected Joint Pain | Mean (SE) | 1.62 (0.07) | 1.08 (0.10) | 0.54 (0.31, 0.78) | < .001 |
| Expected Health | N | 1535 | 398 | | |
| Expected Health | Mean (SE) | 87.86 (0.33) | 87.50 (0.62) | 0.36 (-1.02, 1.74) | 0.610 |
| | | | | | |
| EQ-5D-5L Utility Index | N | 1039 | 547 | | |
| EQ-5D-5L Utility Index | Mean (SE) | 0.80 (0.01) | 0.78 (0.01) | 0.02 (-0.01, 0.04) | 0.15 |
| EQ-5D VAS | N | 1034 | 545 | | |
| EQ-5D VAS | Mean (SE) | 82.19 (0.46) | 79.12 (0.72) | 3.07 (1.40, 4.75) | < .001 |
| Lower Back Pain | N | 1036 | 546 | | |
| Lower Back Pain | Mean (SE) | 2.81 (0.09) | 2.90 (0.13) | -0.09 (-0.41, 0.22) | 0.56 |
| Affected Joint Pain | N | 1030 | 546 | | |
| Affected Joint Pain | Mean (SE) | 1.47 (0.07) | 1.50 (0.10) | -0.03 (-0.26, 0.21) | 0.82 |
| Oxford Hip Score | N | 1031 | 546 | | |
| Oxford Hip Score | Mean (SE) | 41.70 (0.22) | 41.11 (0.35) | 0.59 (-0.22, 1.40) | 0.15 |
| HOOS-12 Pain | N | 873 | 263 | | |
| HOOS-12 Pain | Mean (SE) | 87.57 (0.55) | 86.78 (1.18) | 0.79 (-1.76, 3.33) | 0.55 |
| HOOS-12 Function | N | 873 | 263 | | |
| HOOS-12 Function | Mean (SE) | 88.70 (0.46) | 87.99 (1.00) | 0.71 (-1.44, 2.87) | 0.52 |
| HOOS-12 Quality of Life | N | 873 | 263 | | |
| HOOS-12 Quality of Life | Mean (SE) | 80.67 (0.64) | 80.33 (1.31) | 0.34 (-2.52, 3.20) | 0.82 |
| HOOS-12 Summary | N | 873 | 263 | | |
| HOOS-12 Summary | Mean (SE) | 85.64 (0.50) | 85.03 (1.09) | 0.61 (-1.74, 2.96) | 0.61 |

procedures [14–17]. Better EQ-5D scores have been reported for telephone administration compared to direct patient entry in other populations [18], but was not found in our study.

The addition of telephone call follow up for patients who did not respond electronically, increased data completeness but did not substantially alter the representativeness of the

**Table 5. a: Pre-operative patient-reported outcomes comparing patients responding directly ("electronic") and those responding by telephone ("telephone") for patients undergoing TKA. b: Post-operative patient-reported outcomes comparing patients responding directly ("electronic") and those responding by telephone ("telephone") for patients undergoing TKA.**

| PROM | | Electronic | Telephone | Adjusted Difference (95% CI) | P Value |
|---|---|---|---|---|---|
| EQ-5D-5L Utility Index | N | 2624 | 602 | | |
| EQ-5D-5L Utility Index | Mean (SE) | 0.47 (0.01) | 0.46 (0.01) | 0.01 (-0.02, 0.03) | 0.68 |
| EQ-5D VAS | N | 2591 | 594 | | |
| EQ-5D VAS | Mean (SE) | 69.76 (0.36) | 69.46 (0.68) | 0.30 (-1.22, 1.82) | 0.70 |
| Lower Back Pain | N | 2590 | 594 | | |
| Lower Back Pain | Mean (SE) | 3.32 (0.06) | 3.29 (0.13) | 0.03 (-0.24, 0.30) | 0.83 |
| Affected Joint Pain | N | 2548 | 586 | | |
| Affected Joint Pain | Mean (SE) | 6.70 (0.04) | 6.69 (0.09) | 0.01 (-0.17, 0.20) | 0.88 |
| Oxford Knee Score | N | 2557 | 589 | | |
| Oxford Knee Score | Mean (SE) | 22.38 (0.16) | 22.87 (0.35) | -0.48 (-1.24, 0.27) | 0.21 |
| KOOS-12 Pain | N | 1542 | 434 | | |
| KOOS-12 Pain | Mean (SE) | 40.36 (0.41) | 39.93 (0.82) | 0.43 (-1.37, 2.23) | 0.64 |
| KOOS-12 Function | N | 1535 | 431 | | |
| KOOS-12 Function | Mean (SE) | 46.75 (0.47) | 44.06 (0.97) | 2.69 (0.57, 4.81) | 0.01 |
| KOOS-12 Quality of Life | N | 1530 | 431 | | |
| KOOS-12 Quality of Life | Mean (SE) | 31.99 (0.44) | 30.84 (0.81) | 1.15 (-0.67, 2.97) | 0.22 |
| KOOS-12 Summary | N | 1530 | 431 | | |
| KOOS-12 Summary | Mean (SE) | 39.72 (0.40) | 38.31 (0.79) | 1.41 (-0.32, 3.14) | 0.11 |
| Expected Joint Pain | N | 2539 | 585 | | |
| Expected Joint Pain | Mean (SE) | 2.19 (0.05) | 1.41 (0.09) | 0.78 (0.58, 0.99) | < .001 |
| Expected Health | N | 2578 | 595 | | |
| Expected Health | Mean (SE) | 85.28 (0.30) | 84.94 (0.64) | 0.34 (-1.04, 1.72) | 0.63 |
| | | | | | |
| EQ-5D-5L Utility Index | N | 1564 | 954 | | |
| EQ-5D-5L Utility Index | Mean (SE) | 0.75 (0.01) | 0.78 (0.01) | -0.03 (-0.04, -0.01) | 0.01 |
| EQ-5D VAS | N | 1558 | 952 | | |
| EQ-5D VAS | Mean (SE) | 80.29 (0.39) | 77.54 (0.55) | 2.76 (1.43, 4.08) | < .001 |
| Lower Back Pain | N | 1559 | 951 | | |
| Lower Back Pain | Mean (SE) | 2.77 (0.07) | 2.60 (0.10) | 0.17 (-0.07, 0.41) | 0.18 |
| Affected Joint Pain | N | 1551 | 947 | | |
| Affected Joint Pain | Mean (SE) | 2.43 (0.06) | 2.18 (0.08) | 0.25 (0.05, 0.45) | 0.02 |
| Oxford Knee Score | N | 1554 | 948 | | |
| Oxford Knee Score | Mean (SE) | 37.57 (0.20) | 37.81 (0.27) | -0.24 (-0.90, 0.42) | 0.47 |
| KOOS-12 Pain | N | 1278 | 415 | | |
| KOOS-12 Pain | Mean (SE) | 75.60 (0.54) | 79.38 (0.93) | -3.78 (-5.89, -1.67) | < .001 |
| KOOS-12 Function | N | 1273 | 415 | | |
| KOOS-12 Function | Mean (SE) | 79.91 (0.45) | 81.77 (0.89) | -1.87 (-3.83, 0.10) | 0.06 |
| KOOS-12 Quality of Life | N | 1272 | 415 | | |
| KOOS-12 Quality of Life | Mean (SE) | 70.00 (0.55) | 72.78 (1.04) | -2.79 (-5.10, -0.47) | 0.028 |
| KOOS-12 Summary | N | 1272 | 415 | | |
| KOOS-12 Summary | Mean (SE) | 75.17 (0.47) | 77.98 (0.88) | -2.80 (-4.76, -0.84) | 0.01 |

sample. Although higher completion rates have been considered desirable and minimum proportions have been suggested [3], the representativeness of PROMs samples is rarely reported in research publications. This has important practical and cost implications. Given the high per-patient cost of telephone follow up and annual volume of joint replacement surgery, this

approach is unlikely to be cost-effective on a large scale to improve representativeness but may be used where high rates of follow up are required, e.g., in nested clinical trials. Since the AOANJRR PROMs pilot program, telephone follow up for patients not responding electronically has ceased as part of routine practice. It should be noted that these findings do not allow a direct comparison of primary telephone and electronic follow up, as the telephone follow up was only used for patients who did not or could not respond electronically. Rather, it compares electronic follow up to telephone follow up in people who did not respond electronically.

The strengths of this study are the large sample size and the broad range of institution sizes and locations. There may be differences in representativeness for variables not included in this analysis, for example, socioeconomic factors, ethnicity, education and language proficiency. Another limitation is that the study measured representativeness in patients registered in the PROMs system (i.e., it measured "within-system" representativeness) and did not test the representativeness of the system by looking at patients who were not registered. Therefore, the current study does not address the overall representativeness of the PROMs system and cannot make conclusions about the need to increase the overall coverage of such systems. The findings reported in this paper may not generalise to other clinical registries or jurisdictions.

## Conclusion

Patients undergoing total joint replacement who provide direct electronic PROMs data are younger, healthier and more likely to be female than non-responders, but these differences are small, particularly for post-operative data collection. The addition of telephone call follow up for patients who do not respond electronically increases the response rate but only marginally improves the representativeness of the sample.

## Author Contributions

**Conceptualization:** Ian A. Harris, Yi Peng, Ilana Ackerman, Emma Heath, Stephen E. Graves.

**Data curation:** Michelle Lorimer.

**Formal analysis:** Kara Cashman, Michelle Lorimer, Yi Peng.

**Methodology:** Ian A. Harris, Kara Cashman, Michelle Lorimer, Yi Peng, Ilana Ackerman, Emma Heath.

**Project administration:** Emma Heath, Stephen E. Graves.

**Visualization:** Yi Peng.

**Writing – original draft:** Ian A. Harris.

**Writing – review & editing:** Ian A. Harris, Kara Cashman, Michelle Lorimer, Yi Peng, Ilana Ackerman, Emma Heath, Stephen E. Graves.

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
