## [Decision Letter · Decision Letter 0]

4 Mar 2021

PONE-D-20-33705

Are responders to patient health surveys representative of those invited to participate? An analysis of the Patient-Reported Outcome Measures Pilot from the Australian Orthopaedic Association National Joint Replacement Registry.

PLOS ONE

Dear Dr. Harris,

Thank you for submitting your manuscript to PLOS ONE. After careful consideration, we feel that it has merit but does not fully meet PLOS ONE’s publication criteria as it currently stands. Therefore, we invite you to submit a revised version of the manuscript that addresses the points raised during the review process.

It should be made more clear what the flow of included numbers of patients throughout the analysis is: why were respondents excluded from the total of 18.080 to 11.988 (apart from the 1600 mentioned);  It should also be made clear why fewer than 50% of the THA and TKA patients completed PROMs (differences between tables 1 and 2; and tables 3 and 4). And, more importantly, make sure to interpret your findings in light of any dropout that was not clearly reported, and discuss your results and interpretations critically in light of the small rates of available PROM data (in tables 3 and 4). Discuss why others socio-demographics were not included (namely, education, income, ethnicity), or I would recommend to include these in your analysis. I recommend to further investigate methods to answer research question 2 (improving representativeness), or further explain and argue why the current analyses allow you to answer this research question. Also, interpret your findings in light of the small initial differences found between responders and non-responders. Differences between responders and non-responders are small, but some are significant. As these are due to sample size, the notion of clinically meaningful differences versus statistical significant differences need to be discussed. Consider to rephrase or re-address research question 3, as the two groups are not only different in terms of mode of administration but because they did or did not respond initially/electronically. Clearly try to distinguish throughout the manuscript between mode of administration (electronic vs telephone) and the value of follow-up in itself. Finally, looking at (average) within person differences pre-and post-operation related to responder bias would be a useful addition to the study.

We look forward to receiving your revised manuscript.

Kind regards,

Mathieu F. Janssen, Ph.D.

Academic Editor

PLOS ONE

Journal Requirements:

2. Thank you for stating in your ethics statement in your online submission form "Consent was not obtained for the analyses used in this report as data were analyzed anonymously." Please clarify whether researchers accessed data before or after the data was anonymized.

3. Please include the date(s) on which you accessed the databases or records to obtain the data used in your study.

4. Please include additional information regarding the survey or questionnaire used in the study and ensure that you have provided sufficient details that others could replicate the analyses. For instance, if you developed a questionnaire as part of this study and it is not under a copyright more restrictive than CC-BY, please include a copy, in both the original language and English, as Supporting Information.

Reviewers' comments:

Reviewer's Responses to Questions

**Comments to the Author**

1. Is the manuscript technically sound, and do the data support the conclusions?

Reviewer #1: Partly

Reviewer #2: Yes

2. Has the statistical analysis been performed appropriately and rigorously? 

Reviewer #1: No

Reviewer #2: Yes

3. Have the authors made all data underlying the findings in their manuscript fully available?

Reviewer #1: No

Reviewer #2: No

4. Is the manuscript presented in an intelligible fashion and written in standard English?

Reviewer #1: Yes

Reviewer #2: Yes

5. Review Comments to the Author

Reviewer #1: The abstract should state the response rates with and without telephone follow-up. This is key information to allow readers to interpret the results reported here.

The analysis presented in this paper does not answer research question 2, i.e. whether “adding telephone call follow-up to those responding electronically improves representativeness”. The analyses presented in Tables 1 and 2 do not permit quantifying improvements in representativeness e.g. as a % reduction in bias. Instead, the authors present two separate analyses, both of which show a mismatch between responders (however defined) and non-responders. The reader is left to wonder whether there are improvements in representativeness as a result of adding telephone follow-ups. The statistical literature on matching and standardised mean differences may offer some clues on how to formally test for improvement in representativeness as a result of adding a telephone follow-up. It is also worth noting that there does not appear to be much imbalance between electronic responders and non-responders in the first place and, therefore, there may only be limited scope to improve representativeness?

The phrasing of research question 3 is potentially misleading. As I understand it, the ‘telephone’ group in Tables 3a-4b includes individuals who failed to report data electronically, i.e. telephone follow-up is conditional on non-participation in electronic data collection. The research question 3 (“are patient-reported outcomes different between those who provide information via telephone call and those who provide direct electronic data entry?”) could be misread as comparing two modes of administration in the same population group, for example, to inform clinical registries on whether to design a PROM programme around electronic or telephone mode of data collection. The data in this study cannot answer this more general question since the two populations are systematically different. The telephone group reflects individuals who do not (wish to) take part in electronic data collection, i.e. a conditional sub-group. This needs to be clarified throughout the paper and some of the points made in the discussion may need to be reviewed as well.

More information is required to understand the size of the various samples that are analysed. For example, p.7 explains why 1,600 patients were dropped from the initial sample of 18,080 patients eligible for the PROMs pilot study but readers are not told why the remaining 4,492 patients were excluded to arrive at a final sample of 11,988 patients. It is also unclear why the analysis of post-operative characteristics includes a lower number of patients (3988 + 4211 = 8199). If these discrepancies reflect drop-out then that seems to be much more worrying than the imbalances due to initial non-participation. Finally, Table 2 states that 10,412 patients took part in the pre-operative survey (either electronically or via telephone interview), yet Table 3a reports statistics based on only 1,555 + 399 = 1,984 pre-operative EQ-5D-5L responses. Does this imply that PROMs data were missing for 81.2% of patients participating (i.e. providing some data) in the PROMs pilot study? If so, how meaningful are comparisons based on such a small subsample?

The discussion should acknowledge that any improvements in representativeness (if that can be shown) as a result of telephone follow-up cannot be attributed to the mode of follow-up but could simply arise from having any follow-up that improves overall participation rates.

The discussion of the cost-effectiveness of telephone follow-up (p. 19) focuses on improvement in representativeness as the sole measure of the ‘value’ of such activities. However, there may also be value in having a larger sample size and, consequently, less statistical uncertainty, for example to construct surgeon-specific performance indicators. The conclusion that the authors reach (“this approach is unlikely to be cost-effective on a large scale if it does not improve representativeness”) is only true under a very narrow definition of value. It also suggests that there may be value in more targeted follow-up with underrepresented patient groups.

The discussion should make it clear that the findings reported in this paper may not generalise to other clinical registries or jurisdictions.

p.5 line 75: replace ‘representativeness is biased or is not known’ with ‘sample is unrepresentative’. Sample statistics are not misleading simply because representativeness is not known.

P.7 Line 131. Do the authors mean the critical value chosen to reject the alternative hypothesis?

P.18 line 210: The authors may wish to consider large-scale PROMs data collections in arthroplasty outside of clinical registries. There have been studies examining non-response patterns in hip and knee replacement in the English PROMs programme (e.g. Hutchings et al 2012 in Health and Quality of Life Outcomes). Similar studies may exist for other national PROMs programmes, such as those in Sweden.

p.20 line 244: The authors claim the ‘broad range of demographic […] variables’ as a strength of their study. I would argue that age and sex together do not constitute a broad set of variables. Indeed, I agree with the authors' subsequent statement that other characteristics, such as socioeconomic status and language proficiency but also education and ethnicity are likely to be important. This should be acknowledged as one of the most severe limitations of this study, not a strength.

Reviewer #2: It was a pleasure to review this manuscript. It is well-written and succinct. The study itself appears to be largely descriptive. I only have a few minor and major comments that will hopefully improve the clarity, accuracy, and impact of this analysis.

[Minor] Remove EuroQol from instrument name (EQ-5D-5L is the official name) and "Utility"; should also indicate which country-specific value set was used to calculate the index scores

[Minor] Table 1 and 2: add summary of characteristics for overall cohort to better understand whether there are numerical differences compared to the two subgroups with the overall demographics; if for instance, the differences from the responding group are more similar than the non-responding group... it might provide more to the conclusion that non-response may be less of a concern than previously assumed

[Minor] Table 3a/b: normative to present EQ-5D index scores and their SEs (or utilities, not "utility score") to 2 or 3 decimal places, depending on the published algorithm

[Major] Is race/ethnicity not collected/readily available within this data set? Education level? Marital status? Advance directive status? These demographic characteristics are known to influence PRO measurement. I acknowledge that this is noted in the limitations but seems to be a major limitation that requires further discussion (e.g., which "socioeconomic factors"? How would language proficiency influence PROM measurement?)

[Major] It occurs to me that there is an opportunity to also report not only the mean PROM scores, but the difference in scores between the pre- and post-operative measurements for a subset of patients who provided both. Given the investigators aim to understand the potential selection bias on PROM as a study endpoint, it seems important to capture the potential bias on changes in the patient experience. Please consider whether this is possible with the current analysis and if not, provide justification.

6. PLOS authors have the option to publish the peer review history of their article (what does this mean?). If published, this will include your full peer review and any attached files.

Reviewer #1: No

Reviewer #2: **Yes: **Ernest Law

---

## [Author Response · Author response to Decision Letter 0]

20 Apr 2021

The responses are included in the pdf under "Response to Reviewers", at the end of the pdf

---

## [Decision Letter · Decision Letter 1]

23 Jun 2021

Are responders to patient health surveys representative of those invited to participate? An analysis of the Patient-Reported Outcome Measures Pilot from the Australian Orthopaedic Association National Joint Replacement Registry.

PONE-D-20-33705R1

Dear Dr. Harris,

We’re pleased to inform you that your manuscript has been judged scientifically suitable for publication and will be formally accepted for publication once it meets all outstanding technical requirements.

Kind regards,

Mathieu F. Janssen, Ph.D.

Academic Editor

PLOS ONE

Additional Editor Comments (optional):

Reviewers' comments:

Reviewer's Responses to Questions

**Comments to the Author**

1. If the authors have adequately addressed your comments raised in a previous round of review and you feel that this manuscript is now acceptable for publication, you may indicate that here to bypass the “Comments to the Author” section, enter your conflict of interest statement in the “Confidential to Editor” section, and submit your "Accept" recommendation.

Reviewer #1: All comments have been addressed

2. Is the manuscript technically sound, and do the data support the conclusions?

Reviewer #1: Yes

3. Has the statistical analysis been performed appropriately and rigorously? 

Reviewer #1: Yes

4. Have the authors made all data underlying the findings in their manuscript fully available?

Reviewer #1: No

5. Is the manuscript presented in an intelligible fashion and written in standard English?

Reviewer #1: Yes

6. Review Comments to the Author

Reviewer #1: The revised version is much clearer on which analyses have been performed and how they address the research questions. I have no further queries or concerns.

7. PLOS authors have the option to publish the peer review history of their article (what does this mean?). If published, this will include your full peer review and any attached files.

Reviewer #1: No

---

## [Editor Report · Acceptance letter]

25 Jun 2021

PONE-D-20-33705R1 

Are responders to patient health surveys representative of those invited to participate? An analysis of the Patient-Reported Outcome Measures Pilot from the Australian Orthopaedic Association National Joint Replacement Registry. 

Dear Dr. Harris:

I'm pleased to inform you that your manuscript has been deemed suitable for publication in PLOS ONE. Congratulations! Your manuscript is now with our production department. 

Kind regards, 

on behalf of

Dr. Mathieu F. Janssen 

Academic Editor

PLOS ONE